# Denoising Method Based on Salient Region Recognition for the Spatiotemporal Event Stream

**DOI:** 10.3390/s23156655

**Published:** 2023-07-25

**Authors:** Sichao Tang, Hengyi Lv, Yuchen Zhao, Yang Feng, Hailong Liu, Guoling Bi

**Affiliations:** 1Changchun Institute of Optics, Fine Mechanics and Physics, Chinese Academy of Sciences, Changchun 130033, China; tangsichao21@mails.ucas.ac.cn (S.T.); zhaoyuchen@ciomp.ac.cn (Y.Z.); fenyang16@mails.ucas.edu.cn (Y.F.); liuhailong@ciomp.ac.cn (H.L.); biguoling@ciomp.ac.cn (G.B.); 2College of Materials Science and Opto-Electronic Technology, University of Chinese Academy of Sciences, Beijing 100049, China

**Keywords:** event cameras, denoising methods, salient region recognition

## Abstract

Event cameras are the emerging bio-mimetic sensors with microsecond-level responsiveness in recent years, also known as dynamic vision sensors. Due to the inherent sensitivity of event camera hardware to light sources and interference from various external factors, various types of noises are inevitably present in the camera’s output results. This noise can degrade the camera’s perception of events and the performance of algorithms for processing event streams. Moreover, since the output of event cameras is in the form of address-event representation, efficient denoising methods for traditional frame images are no longer applicable in this case. Most existing denoising methods for event cameras target background activity noise and sometimes remove real events as noise. Furthermore, these methods are ineffective in handling noise generated by high-frequency flickering light sources and changes in diffused light reflection. To address these issues, we propose an event stream denoising method based on salient region recognition in this paper. This method can effectively remove conventional background activity noise as well as irregular noise caused by diffuse reflection and flickering light source changes without significantly losing real events. Additionally, we introduce an evaluation metric that can be used to assess the noise removal efficacy and the preservation of real events for various denoising methods.

## 1. Introduction

### 1.1. Event Cameras and Current Status Introduction

Image sensor technology has rapidly developed in modern society, with mainstream devices like CCD and CMOS providing intuitive and pleasing images [1,2,3]. However, frame-based sensors generate a large amount of redundant information, and due to frame rate limitations, information from high-speed objects may be lost. To accommodate more complex needs, the frame rate of frame-based cameras has to be set at its upper limit. Therefore, the dynamic vision sensor based on neural morphology loaded in the event camera has been invented. This biomimetic visual device is inspired by the structure of biological retinas [4,5,6,7,8,9], and its detailed structure is shown in Figure 1.

Each pixel of this sensor contains a fast logarithmic photosensitive circuit, a differential amplifier circuit, and a comparator. These circuits function similarly to cone cells, bipolar cells, and ganglion cells on the retina. Specifically, the fast logarithmic photosensitive circuit is used to convert light signals into electrical signals, the differential amplifier circuit can amplify the change in light intensity, and the comparator, similar to ganglion cells, is used to induce changes in the light intensity signal. When the change in light intensity exceeds the threshold, it outputs an ON signal; otherwise, it outputs an OFF signal. The process of a single-pixel generating events is illustrated in Figure 2 [10], and the schematic diagram of the entire event stream generation process is shown in Figure 3.

It is worth noting that due to its unique pixel structure, just like human attention mechanisms, dynamic vision sensors only respond to areas where the light intensity changes in the scene, eliminating data redundancy and making them widely applicable in high-speed image processing and machine vision applications. The output ON/OFF signal is called event $e_{i}$ = e($x_{i},y_{i},t_{i},p_{i}$), which contains position, microsecond timestamp, and polarity information. Combining the characteristics of dynamic vision sensors, we call the output event as spatio-temporal event streams. Spatio-temporal event streams can be defined as:(1)$$E=\sum_{i=0}^{n-1}e(x_{i},y_{i},t_{i},p_{i}),$$
where e is an event in the spatio-temporal event stream; [x, y] represents the position of the pixel that generated the event; p ∈ {0,1} indicates the polarity of the light change at the pixel causing the event; t represents the time of the event occurrence; i is the index of the event in the spatio-temporal event stream; and Σ represents adding a new event to the spatio-temporal event stream.

Such characteristics enable event cameras loaded with dynamic vision sensors to support pixel-level parallel signal processing and event-driven readout. Compared to conventional frame image cameras, they have advantages such as low power consumption, high temporal resolution, high dynamic range, no motion blur, and small data volume. These features allow event cameras to provide new development areas for visual applications and be used to perform some challenging tasks such as target tracking [11,12,13], target recognition and surveillance [14,15,16,17,18], depth estimation [19], optical flow estimation [20], high dynamic range image reconstruction [21], segmentation [22], guidance [23,24], simultaneous localization and mapping [25], and other fields.

However, at the same time, event cameras are also susceptible to various interferences from the external environment, resulting in noisy events. These noises are antagonistic to the real events in the event stream, and their presence affects image quality, occupies unnecessary bandwidth, and poses significant difficulties for algorithm development, subsequent data processing and backend applications.

### 1.2. Existing Denoising Methods

Various factors contribute to the noise generated by event cameras. Under constant lighting conditions, event cameras may erroneously output events due to threshold fluctuations, charge injection effects, and junction leakage currents [8,26,27], even if the perceived brightness of the pixels does not change or the change does not reach the threshold. These isolated, unconnected events that do not reflect the target information are called background activity noise events. Background activity events differ from real activity events in that they lack temporal correlation with newly arriving events in the spatial neighborhood, while real activity events show meaningful correlations.

In order to remove these noises, Ojeda et al. [28] combined a novel point process filter with an adaptive time window scheme that can improve classification accuracy with a binary neural network that allows hardware to work more efficiently and implemented the entire denoising system using field-programmable gate arrays (FPGA). Li et al. [29] proposed a denoising method with pixel-parallel noise and spatial redundancy suppression capabilities, which can remove irrelevant events to some extent, but this method cannot effectively solve background activity events generated under low light conditions. Delbruck et al. [30] proposed a spatio-temporal filter that classifies events with fewer than eight in the spatio-temporal neighborhood as noise. The drawback of this method is that when two background noises are close enough in a spatio-temporal region, they will be recognized as real events by the algorithm. Liu et al. [26] proposed a filter that reduces memory requirements by grouping pixels, where memory units store subsampled pixel groups rather than projecting each pixel separately into a memory unit. However, the subsampling factor affects the denoising accuracy of the filter. When the subsampling factor is greater than two, the denoising accuracy decreases. Khodamorad and Kastner [31] reduced the memory complexity of the filter from O(N^2^) to O(N) and proposed a noise filter that can use less memory space to store events and their timestamps, greatly reducing the cost of memory required. The principle of this method is that the most recent event and its corresponding polarity and timestamp in each row and column of the event stream are stored in two 32-bit memory units. Within a short time window, if two events are captured in the same column but different rows, the most recent event will overwrite the old event in memory.

The denoising methods mentioned above are mainly used to deal with background activity noise, with relatively limited functionality and mediocre performance. In order to cope with more application scenarios, researchers have developed denoising methods with a wider range of applications. This method is mainly divided into two categories: the first category of algorithms consider event density as the main influencing factor in the denoising process, while the other category of algorithms consider events that do not conform to the ideal instantaneous log intensity change as noise events. The most widely used filtering method among these is the nearest neighbor-based filtering based on spatio-temporal correlation [26,30,32,33,34]. In this filter, the attributes of previously generated events within the spatio-temporal neighborhood are used to determine whether the newly arrived events represent real activity. The parameters of the spatio-temporal window need to be adjusted by the user. Therefore, this method requires additional storage resources to retain the attributes of previous and newly arrived events for processing. Afshar et al. [35] also used the concept of density in their method, using the average event density to control the size of the judgment threshold. However, implementing the significantly exponential decay neighborhood mentioned in their method would be costly. Feng et al. [36] proposed a density matrix, projecting each arriving event into its spatio-temporal region, and estimating the threshold for judging random noise and hot pixels from the static parts of each scene. Wu et al. [10] proposed a method based on probabilistic undirected graph models and iterative conditional mode for sorting signal and noise events. They then manually designed an energy function and used iterative algorithms to perform the energy minimization process, obtaining highly correlated real events and removing less correlated noise events. However, due to the local optimality of the ICM algorithm and the long and uncertain runtime of the program, this method is not suitable for real-time denoising. Guo et al. [37] used memory proportional to the number of pixels to improve the accuracy of the correlation check step in the algorithm. Compared to existing methods, their STCF algorithm filters out events that lack k past events to support them as real events rather than as noise events in the correlation time τ, not just one. Wang et al. [38] proposed an improved linear comb filter to remove event flickering, using both delay feedback and delay feedforward to obtain the delayed version response of the original signal, effectively passing the DC component of the signal while strongly attenuating all harmonics of the desired fundamental frequency. Since the filter is based on a linear filter design, there is a brief transient when the internal filter state converges to attenuate undesired signals. This algorithm allows parallel pixel-wise computation and is the first to effectively remove flickering events, making it suitable for implementation on low-latency and low-power hardware circuits such as FPGA.

The performance of the existing denoising methods largely depends on adjustable parameters, such as spatio-temporal window size, event camera settings, pixel bandwidth and threshold, ambient lighting conditions, and camera motion [31,36,39,40,41]. These parameters are related to the application algorithm, and manually adjusting them may yield satisfactory denoising results, particularly under good lighting conditions. However, the control operation also generates noise and reduces the speed and sensitivity of data processing.

Moreover, most of the aforementioned denoising methods often fail to handle irregular events such as light ray diffusions or light source flickers, as some of them are real events that are difficult to remove, and forcibly removing them may lead to the deletion of real events that have similar spatio-temporal features. However, if these events are considered real, it may undermine the advantages of event cameras. This can be seen in Scene B in the real-world experimental phase in Section 3.2. When the environmental light changes abruptly, the APS channel that captures frame images will lose the image information of the pixels near the light source due to overexposure. The DVS channel, on the other hand, can accurately capture the event information occurring on those pixel positions under such conditions, except that these events are mixed with a large number of noise events and are not easily distinguishable. Moreover, since the event camera only responds to pixel positions where the brightness changes exceed a threshold, it itself cannot perceive the static image information that a frame image camera can perceive.

So it is important to filter out diffuse reflection events, high-frequency flicker events, etc., while preserving real events as much as possible to better utilize the advantages of event cameras. One of the future development trends of event cameras is to form a multi-modal visual information processing system in conjunction with other sensors, such as frame image cameras and infrared sensors, leveraging the strengths and overcoming the weaknesses, eliminating useless interference information, and combining the remaining useful information to achieve comprehensive perception of the shooting environment.

Hence, there is an urgent need for a new type of dynamic denoising method based on event streams, which dynamically adjusts parameter thresholds according to different situations, and while effectively preserving real events in the event stream, removes as much regular and irregular noise as possible. Our method can achieve this effect.

### 1.3. The Main Contribution of This Article

In this paper, we propose a new event stream denoising method based on salient region recognition, designed as follows. First, the motion of an object generates a large number of events, which we call real events. At the same time, the event camera hardware itself and environmental factors produce random noise events. It can be seen that the event stream L output by the event camera is a mixture of real events and noise events. Based on this situation, we propose the Main Points Area Recognition algorithm. In this algorithm, we combine the improved Pearson correlation formula and Spearman correlation formula to obtain the main event stream U, which is primarily composed of real events, and the non-main event stream N, which is dominated by noise events. This step classifies events based on their overall relevance to the event stream, identifying some noise events with strong self-correlation, such as diffuse reflection noise.

Next, we need to further identify misjudged real events and noise events in the two event streams. We define the joint probability distribution P(U, L) and P(N, L) that can represent the correlation between events based on the Markov Random Field and then set a reasonable potential function to convert the probability size problem into an energy size problem. Based on the relationship that the joint probability distribution can be represented as the product of the function on its maximum clique, we define the energy function E(U, L) and E(N, L) that can describe the relationship between different events after energy conversion. The smaller the value of this energy function, the smaller the energy and the higher the probability, indicating that the correlation between the two events is stronger. The threshold for judging whether an event is noise is different in the main event stream and the non-main event stream. Based on this situation, we propose the Event-Energetic Iterated Modes algorithm. In this algorithm, we use the improved Manhattan distance definition, which reflects both temporal and spatial characteristics according to the characteristics of the event stream, as the adaptive weight value for judging event categories and correlation strength, and finally obtain the real events and noise events in the two event streams. The entire processing stream is shown in Figure 4.

Our contributions mainly include the following four parts:We propose the Main Points Area Recognition (MPAR) algorithm that can divide the event stream into the main event stream and the non-main event stream according to the overall correlation. This can identify unconventional noise like the diffuse reflection noise in the event stream that traditional denoising methods cannot effectively handle;We establish a probability model that reflects the correlation between events based on the Markov Random Field (MRF) and establish a corresponding energy function to reflect the probability of an event being noise through the energy size;We propose the Event-Energetic Iterated Modes (EEIMs) algorithm that uses an adaptive weight threshold as the basis for judging whether an event in different energized event streams is noise. This allows the use of different discrimination standards for the two event streams, maximizing the removal of noise events while retaining real events as completely as possible;We propose a new evaluation criterion that can be used to assess the noise removal effect in the event stream after denoising, as well as the retention effect of real events. Compared to the traditional ones that evaluate the denoising effect through results, such as using signal-to-noise ratio, our method can more accurately reflect the performance of a denoising algorithm.

## 2. Materials and Methods

### 2.1. Main Points Area Recognition Algorithm

Nowadays, most denoising algorithms use the same basis for correlation judgment in the real event area mixed with noise events and the noise event area mixed with real events. For example, if there are k supporting events within a neighboring area of an event within a relevant time τ, then the event is considered a real event; otherwise, it is considered a noise event. Alternatively, if the real/noise event ratio around the mapping structure where the event is located is k, then the event is considered a real/noise event. In this case, the mentioned k value is an invariant parameter throughout the process. At the same time, methods that use fixed spatio-temporal windows, discriminant thresholds, and other adjustable parameters to perform event-type judgments are prone to missing unconventional noise with strong self-correlation, such as diffuse reflection noise or flickering noise. In other words, these methods may output unconventional noise as real events during denoising. Based on this situation, we use the MPAR algorithm to divide the original event stream L into the main event stream U, which is mainly composed of real events, and the non-main event stream N, which is mainly composed of noise events. On one hand, this identifies unconventional noise events with strong self-correlations, and on the other hand, it allows subsequent algorithms to use dynamic judgment thresholds based on the different characteristics of the two event streams for further denoising operations.

When judging and dividing event stream regions, we propose the improved Pearson correlation coefficients $\rho_{1}$ and Spearman correlation coefficients $\rho_{2}$ based on the knowledge of Pearson correlation and Spearman correlation and according to the characteristics of the event stream. The formulas are as follows:(2)$$\rho_{1}=\frac{cov(e_{i}.x,e_{i}.y)}{\left|e_{\left(i-1\right)}.x-e_{i}.x|\cdot |e_{\left(i-1\right)}.y-e_{i}.y\right|},$$
(3)$$\rho_{2}=\frac{\frac{1}{i}\sum_{m=1}^{i}\left[(e_{m}.x-e_{m}.\bar{x})\cdot (e_{m}.y-e_{m}.\bar{y})\right]}{\sqrt{\left[\frac{1}{i}{\sum_{m=1}^{i}\left(e_{m}.x-e_{m}.\bar{x}\right)}^{2}]\cdot {\left[\frac{1}{i}\sum_{m=1}^{i}(e_{m}.y-e_{m}.\bar{y}\right)}^{2}\right]}},$$
where $e_{i}.x$ represents the x-coordinate of the i-th event; $e_{i}.y$ represents the y-coordinate of the i-th event; and $e_{i}.\bar{x}$ and $e_{i}.\bar{y}$ represent the weighted averages of the x and y coordinates of other events at the time node where the i-th event occurs, with the frequency of event occurrence as the weight, respectively.

The Pearson correlation coefficient, also known as the product-moment correlation coefficient, is used to measure the linear association and normal distribution association between two random variables. Its range is between [−1, 1], where −1 represents a completely negative correlation, 1 represents a completely positive correlation, and 0 represents no correlation. In other words, when the coefficient is 0, the changes in the two variables have no effect on each other. The Spearman correlation coefficient belongs to the rank correlation coefficient and can reflect the correlation between the changing trends of the two random variables, similar to the Pearson correlation coefficient. However, the Spearman correlation coefficient measures the correlation between non-linear relationship variables. When the $\rho_{1}$ and $\rho_{2}$ values corresponding to the two correlation coefficients are between 0.8 and 1, the correlation between the corresponding events is strong. These two improved correlation coefficient formulas can fully measure the correlation between events in the event stream, thus dividing the original event stream L into the main event stream U, which is mainly composed of real events, and the non-main event stream N, which is mainly composed of noise events. By combining the above-related knowledge, we can write an algorithm to determine whether an event belongs to the main event stream (Algorithm 1).
**Algorithm 1:** Main Points Area RecognitionInput: The spatio-temporal event stream $\sum_{i=0}^{n-1}e(x_{i},y_{i},t_{i},p_{i})$Output: The main event stream $\sum_{i=0}^{n-1}e_{T}(x_{i},y_{i},t_{i},p_{i})$ and the non-main event stream $\sum_{i=0}^{n-1}e_{F}(x_{i},y_{i},t_{i},p_{i})$1  For $e_{\left(x_{i},y_{i},t_{i},p_{i}\right)}$ in $\sum_{i=0}^{n-1}e(x_{i},y_{i},t_{i},p_{i})$2         Calculate the hierarchical weight $q_{1}=\sum_{m=1}^{i}\left[{\left(e_{m}.x\right)}^{n-1}e^{-(e_{m}.x)}\right]$3         Calculate the hierarchical weight $q_{2}=\sum_{m=1}^{i}\left[{\left(e_{m}.y\right)}^{n-1}e^{-(e_{m}.y)}\right]$4         Use the Formulas (2) and (3) to calculate the $\rho_{1}$ and $\rho_{2}$5         If ($\rho_{1}>0.5+1/\sqrt[{i}]{\log_{10}{{(q}_{1}\cdot q}_{2})}$|$\rho_{2}>0.5+1/\sqrt[{i}]{\log_{10}{{(q}_{1}\cdot q}_{2})}$)do6                  Include this event in the main event stream array container7         Else8                  Include this event in the non-main event stream array container9  Output new streams $\sum_{i=0}^{n-1}e_{T}(x_{i},y_{i},t_{i},p_{i})$ and $\sum_{i=0}^{n-1}e_{F}(x_{i},y_{i},t_{i},p_{i})$10  End

### 2.2. The Event Energy Model Based on Markov Random Field

The core concept of the Markov Random Field is the Markov property, which indicates that the value of the current observed variable is influenced by the current state and other states. To elaborate, it means that after a sequence of random variable states is unfolded in a chronological order, the conditional probability distribution of its future states, given the current state and all past states, depends only on the current state. In other words, given the current state of the random variable, its value is independent of past states, and such a random process is said to have the Markov property. The term “random field” refers to the configuration when a value from a certain space is randomly assigned to each position according to a certain distribution, the entirety of which is called a random field. The combination of these two concepts forms the Markov Random Field.

The model based on the concept of Markov Random Field [42] is composed of nodes and connecting lines, and it is used to describe the undirected interactions between variables. As shown in Figure 5, nodes represent variables, and the connecting lines represent the probability of interaction between neighboring variables. Based on the above prior knowledge, we utilize the Markov Random Field model and the inherent correlation between the event camera event stream, using the nodes in the model to represent the events in the event stream, and the connecting lines to represent the correlation between the events.

In the previous section, we initially divided the event stream into two based on the correlation of the events with the overall event stream: the main event stream $\sum_{i=0}^{n-1}e_{U}(x_{i},y_{i},t_{i},p_{i})$, primarily composed of real events, and the non-main event stream $\sum_{i=0}^{n-1}e_{N}(x_{i},y_{i},t_{i},p_{i})$, primarily composed of noise events. The main event stream U is generated by object motion, while the non-main event stream N is generated by inherent hardware and external interference. The combination of these two event streams forms the original event stream L,
(4)L=U+N.

However, to further filter out the noise events in the main event stream and the real events within the non-main event stream, it is necessary to combine the characteristics of the two event streams and design a distinction method that can dynamically adjust according to the situation and better reflect the correlation between events.

In the vast majority of cases, the event stream is dominated by real events, implying that there is a strong correlation between the events in event stream U and event stream L, whereas the correlation between the events in event stream N and event stream L is relatively smaller. Based on this situation, we can combine the related properties of the Markov Random Field model to represent the relationships between the events in event stream U and event stream L, as well as between the events in event stream N and event stream L, using joint probability distribution, and then proceed to the next step of processing. According to the Hammersley–Clifford theorem, the joint probability distribution of the Markov Random Field model can be represented as the product form of non-negative functions on its maximum clique of random variables. This operation is also known as the factor decomposition of the Markov Random Field model,
(5)$$P(U,L)=\frac{1}{Z}\prod_{C}\psi_{C}(U_{C},L_{C}),$$

C is the maximal clique in the model. In the Markov Random Field model, any subset of nodes that are all connected by edges is called a clique. If C is a clique in the model and no additional node can be added to form a larger clique, then C is referred to as a maximal clique.

The potential function $\psi_{C}(U_{C},L_{C})$ is defined in units of maximal cliques, i.e., a graph with several maximal cliques will have several potential functions. The potential functions are used to quantify the joint probability of random variables in the maximal clique, but they are not normalized. Since the potential function is restricted to be strictly greater than zero to ensure P(U,L)≥ 0, it is more convenient to represent the potential function in exponential form,
(6)$$\psi_{C}(U_{C},L_{C})=e^{-E(U_{C},L_{C})},$$

Z is the partition function, also known as the normalization factor, which corresponds to the sum of the factorization results of all maximal cliques in the Markov Random Field model, used to normalize the product form of P(U,L) to a probability form,
(7)$$Z=\sum_{UL}\prod_{C}\psi_{C}(U_{C},L_{C}),$$

$E(U_{C},L_{C})$ represents the energy function within the corresponding clique. We combine the two equations above to represent the potential function in the joint probability distribution with the energy function. At this point, the joint probability distribution is in the form of a Gibbs distribution,
(8)$$P(U,L)=\frac{1}{Z}\prod_{C}e^{-E(U_{C},L_{C})}=\frac{1}{Z}e^{-E(U,L)},$$

E(U,L) is the total energy function corresponding to the Markov Random Field model of event stream U and event stream L,
(9)$$E(U,L)=\sum_{C}E(U_{C},L_{C}).$$

In the Markov Random Field model that we use, $U_{x,y,t}$ represents a single event in U, $N_{x,y,t}$ represents a single event in N, and $L_{x,y,t}$ represents a single event in L. In the event stream, pixels only exist in three states: an event with increased brightness occurs ($p_{i}$ = 1); an event with decreased brightness occurs ($p_{i}$ = 0); and no event occurs. Based on this characteristic, we can define $U_{x,y,t}\in \{-1,0,1\}$, $N_{x,y,t}\in \{-1,0,1\}$, and $L_{x,y,t}\in \{-1,0,1\}$, where x,y,t, respectively represent the horizontal position information of the event, vertical position information and the timestamp information.

Moreover, when $U_{x,y,t}$ is a real event, it has a strong correlation with $L_{x,y,t}$. In addition, $U_{x,y,t}$ has a strong correlation with other events $U_{x\pm ∆x,y\pm ∆y,t}$ and $U_{x\pm ∆x,y\pm ∆y,t\pm ∆t}$ in the spatial and temporal neighborhoods. However, when $U_{x,y,t}$ is a noise event in U, the situation is completely opposite; at this time, $U_{x,y,t}$ does not have a strong association with $L_{x,y,t}$, $U_{x\pm ∆x,y\pm ∆y,t}$, and $U_{x\pm ∆x,y\pm ∆y,t\pm ∆t}$. ∆x, ∆y and ∆t,respectively, represent the differences in horizontal position, vertical position, and timestamp between the original event and other events in the neighborhood of this event.

According to the above knowledge, the Markov Random Field model of event stream U and event stream L includes the following four maximal cliques: $\left\{U_{x,y,t},L_{x,y,t}\right\}$, $\left\{U_{x,y,t},U_{x\pm ∆x,y\pm ∆y,t}\right\}$, $\left\{U_{x,y,t},U_{x,y,t\pm ∆t}\right\}$, and $\left\{U_{x,y,t},U_{x\pm ∆x,y\pm ∆y,t\pm ∆t}\right\}$. For maximal clique $\left\{U_{x,y,t},L_{x,y,t}\right\}$, in order to describe the relationship between events in the clique, we defined the corresponding energy function,
(10)$$E_{UL}(U_{C},L_{C})=e^{-(\sum_{C}x_{U}+\sum_{C}y_{U}+\sum_{C}t_{U})}(||U_{x,y,t}-L_{x,y,t}|{|}_{0}+1).$$

In this energy function, when $U_{x,y,t}$ and $L_{x,y,t}$ are in the same state, the energy produced by the energy function is lower, and the corresponding correlation is higher. When $U_{x,y,t}$ and $L_{x,y,t}$ are in opposite states, the energy produced by the energy function is higher, and the corresponding correlation is lower.

Similarly, we also defined corresponding energy functions for the other three maximal cliques,
(11)$$E_{UV}(U_{C},L_{C})=\frac{e^{-(\sum_{C}x_{U}+\sum_{C}y_{U}+\sum_{C}t_{U})}}{\alpha }(||U_{x,y,t}-U_{x\pm ∆x,y\pm ∆y,t}|{|}_{0}+1),$$
(12)$$E_{UT}(U_{C},L_{C})=\frac{e^{-(\sum_{C}x_{U}+\sum_{C}y_{U}+\sum_{C}t_{U})}}{\beta }(||U_{x,y,t}-U_{x,y,t\pm ∆t}|{|}_{0}+1),$$
(13)$$E_{UVT}(U_{C},L_{C})=\frac{e^{-(\sum_{C}x_{U}+\sum_{C}y_{U}+\sum_{C}t_{U})}}{\left(\alpha +\beta \right)}(||U_{x,y,t}-U_{x\pm ∆x,y\pm ∆y,t\pm ∆t}|{|}_{0}+1),$$

α and β are non-negative dynamic weight parameters within the corresponding maximal cliques. α reflects the correlation of events in space,
(14)$$\alpha =\frac{\pi (\delta x+\delta y)}{4(M_{x}+M_{y})},$$

β reflects the correlation of events in time,
(15)$$\beta =\frac{∆t_{min}}{∆t_{max}},$$

δx and δy,respectively, represent the pixel differences in the horizontal and vertical directions of the two events in the energy function. $M_{x}$ and $M_{y}$ represent the maximum differences in both the horizontal and vertical directions among all events within all maximum cliques, respectively. $∆t_{min}$ represents the minimum time interval between the $U_{x,y,t}$ event and other events within all maximum cliques, while $∆t_{max}$ represents the maximum time interval.

In summary, we can obtain the complete energy function E(U,L) corresponding to the main event stream U and the original event stream L,
(16)$$E(U,L)=\sum_{C_{UL}}E_{UL}(U_{C},L_{C})+\sum_{C_{UV}}E_{UV}(U_{C},L_{C})+\sum_{C_{UT}}E_{UT}(U_{C},L_{C})+\sum_{C_{UVT}}E_{UVT}(U_{C},L_{C}),$$

$C_{UL}$, $C_{UV}$, $C_{UT}$, and $C_{UVT}$ correspond to the four maximal cliques $\{U_{x,y,t},L_{x,y,t}\},$ $\left\{U_{x,y,t},U_{x\pm ∆x,y\pm ∆y,t}\right\}$, $\left\{U_{x,y,t},U_{x,y,t\pm ∆t}\right\}$, and $\left\{U_{x,y,t},U_{x\pm ∆x,y\pm ∆y,t\pm ∆t}\right\}$, respectively. The maximal clique a represents the relationship between events in event stream U and events in event stream L, and the other three maximal cliques represent the relationship between events in the spatial-temporal neighborhood in event stream U.

Similarly, we can obtain the complete energy function E(N,L) corresponding to the non-main event stream N and the original event stream L,
(17)$$E(N,L)=\sum_{C_{NL}}E_{NL}(N_{C},L_{C})+\sum_{C_{NV}}E_{NV}(N_{C},L_{C})+\sum_{C_{NT}}E_{NT}(N_{C},L_{C})+\sum_{C_{NVT}}E_{NVT}(N_{C},L_{C}),$$

Combining the relationships between the aforementioned energy functions, we can derive the event energy model based on Markov Random Field. And as shown in Figure 6, it shows the relationship between the maximum cliques. However, the event situation in the noise-dominated event stream N is different from that in event stream U. In the previous section, when we distinguished the main event stream U and the non-main event stream N based on overall correlation, random noise events such as background activity noise with low intercorrelation and unconventional noise events like diffusive reflection noise with high intercorrelation were both categorized into the non-main event stream N. For diffusive noise in the non-main event stream, although there is a strong correlation among the events within this noise region, they do not strongly correlate with the events in the original event stream L. In contrast, the random noise events in event stream N have a weak correlation both with other events in N and with events in stream L. However, the real events in event stream N have strong correlations with both N and L events. With these characteristics as a judgment basis, we can make a good distinction among the conventional noise events, unconventional noise events, and real events in the event stream N.

After representing the relationships between events using energy functions in this section, we can more effectively measure the correlation between events within the same or different event streams. Moreover, based on the different event characteristics corresponding to the energy functions of the two event streams obtained in this section, we can use the EEIM algorithm in the next section, which includes dynamic thresholds belonging to different event streams, to further identify the real events in the non-main event stream and the noise events in the main event stream. In this way, we can effectively preserve the real events while removing both conventional and unconventional noise events.

### 2.3. Event-Energetic Iterated Modes Algorithm

Nowadays, there are many methods to deal with energy-processed event streams. The most recent method is the ICM algorithm used by Wu et al. [10]. However, the event stream processed with this algorithm still has many residual noise events, because the ICM algorithm uses a strategy similar to greed, making it prone to a local optimum. If we want to achieve a better denoising result, we need to adopt a strategy that can achieve global optimum.

A better choice is to use the simulated annealing algorithm. Simply put, the principle of the simulated annealing algorithm is to start from a higher energy, which is called the initial energy. As the energy parameters continue to decrease, the results in the algorithm tend to stabilize. But the stable solution obtained this way is still a local optimal solution. At this time, according to the Metropolis criterion, the simulated annealing algorithm will jump out of such a local optimal solution with a certain probability to find the global optimal solution of the objective function. However, such processing is still full of uncertainty, and the results are random each time. This makes the denoising effect unstable and also increases the computational resource occupancy of the algorithm.

Therefore, we need to set an adaptive dynamic energy threshold, which can adjust itself in real time according to the constantly changing characteristics of the event stream, so as to make a precise qualitative division of the probability of whether an event is a noise event or a real event. On the other hand, it can effectively avoid the situation of constantly pursuing energy minimization, thus leading to local optimum. According to the complete energy function and its expression set in the previous section, we use the improved formula of Manhattan distance definition to calculate the adaptive dynamic energy threshold EnergyValve we need. Then, we use the time and space information after the discrete Hilbert transform to obtain the dynamic weight value freq, so as to reduce or increase the effective weight of the energy function calculation result. In this way, we not only include time information and space information as judgment basis, corresponding to the components in the complete energy function, but also allow the energy function to deal with the differences in correlation between the different event streams, thus balancing the denoising ability of the EEIM algorithm and greatly improving the algorithm’s compatibility with different types of noise events. The process of the EEIM algorithm is as follows (Algorithm 2).
**Algorithm 2:** Event-Energetic Iterated ModesInput: The main event stream $\sum_{i=0}^{n-1}e_{T}(x_{i},y_{i},t_{i},p_{i})$ and the non-main event stream $\sum_{i=0}^{n-1}e_{F}(x_{i},y_{i},t_{i},p_{i})$Output: The event stream after denoising completion $\sum_{i=0}^{n-1}e_{new}(x_{i},y_{i},t_{i},p_{i})$1  For ${e_{T}}_{\left(x_{i},y_{i},t_{i},p_{i}\right)}$ in $\sum_{i=0}^{n-1}e_{U}(x_{i},y_{i},t_{i},p_{i})$2         Use the formula $\hat{e[i]}=\frac{2}{\pi }\sum_{m=-\infty }^{+\infty }e[i-m]\frac{\sin^{2}\left(m\pi /2\right)}{m}$ to perform a Hilbert discrete transformation on the x-axis, y-axis, and timestamp information3         Calculate the derivative values $o_{i}$.u,$o_{i}$.v,$o_{i}$.w of the results in step 24         Use the formula $K=\frac{C^{\prime \prime }}{(1+C^{\prime 2}{)}^{\frac{3}{2}}}$ to calculate the curvature α,β,γ corresponding to $o_{i}$.u,$o_{i}$.v,$o_{i}$.w5         Calculate $freq=\alpha \log_{2}|\frac{1}{o_{i}.u}|+\beta \log_{2}\left|\frac{1}{o_{i}.v}\right|+\gamma \log_{2}|\frac{1}{o_{i}.w}|$6         Calculate $EnergyValve={|o}_{\left(i-1\right)}.u-o_{i}.u|+|o_{\left(i-1\right)}.v-o_{i}.v|+|o_{\left(i-1\right)}.w-o_{i}.w|$7         If ($\frac{E(U,L)}{2\sqrt[{3}]{freq}}$ > EnergyValve) do8                   Mark the corresponding event data as noise events9  For ${e_{F}}_{\left(x_{i},y_{i},t_{i},p_{i}\right)}$ in $\sum_{i=0}^{n-1}e_{F}(x_{i},y_{i},t_{i},p_{i})$10         Repeat steps 2, 3, 4, 5, 611         If ($\frac{E(N,L)}{2\sqrt[{3}]{freq}}$ < EnergyValve) do12                   Mark the corresponding event data as real events13  Extract new data $e_{new}(x_{i},y_{i},t_{i},p_{i})$ from original data14  Output the event stream $\sum_{i=0}^{n-1}e_{new}(x_{i},y_{i},t_{i},p_{i})$15  End


## 3. Results

In this section, we divide the experimental process into two stages: denoising experiments in simulated scenarios and denoising experiments in real scenarios. Each stage uses different evaluation system methods to compare the noise reduction effects of various algorithms. The simulated event stream dataset used in this process is derived from high-frame-rate videos shot with a high-speed camera, which we converted using an event camera simulator [43]. The real event stream dataset was obtained by shooting with an event camera.

The experimental equipment used in this experiment includes a DAVIS346 event camera and a 240-frame high-frame video recording device. The DAVIS346 event camera includes two data output channels: the APS (active pixel sensor) channel that outputs traditional frame image information, and the DVS channel that outputs event stream information. The algorithm part involved in the experiment is implemented on the MATLAB 2022b platform. The computer running the algorithm program is a laptop equipped with an Intel(R) Core(TM) i7-11800H @2.30GHz processor, 16G RAM, and NVIDIA GeForce RTX 3060 6GB. The operating system of the computer is Windows 11.

When conducting comparative experiments, the algorithms we chose for comparison are the most widely used Khodamoradi [31] denoising algorithm, Yang [36] denoising algorithm, and the latest Guo [37] denoising algorithm STCF. The time window of the Khodamoradi denoising algorithm is set to 1 ms. The parameters of the Yang denoising algorithm are set to the default values in [36], that is, the time window is set to 5 ms, the spatial window is 5 × 5 pixels, and the density is three. As for the parameters of the Guo denoising algorithm, we set the size of the number of correlated neighbors to four here.

### 3.1. Denoising Experiment in Simulated Scenarios

In this experimental stage, we first use a high-frame video shooting device to record a piece of material at a fixed point, and then use an event camera simulator to convert the high-frame video shot into an event stream. As shown in Figure 7, the content of the shooting is a short-distance walk of a single pedestrian in a fixed scene. Since the target is single and the scene is fixed, the simulated event stream can be regarded as a noise-free event stream data.

According to the research results of the Khodamoradi team [31], it can be understood that the background activity noise that accounts for the largest proportion in the event stream noise follows a Poisson distribution. Therefore, based on the proportion of the simulated event stream quantity (10–100%), we inject a certain amount of noise that follows a Poisson distribution into the original event stream. Then, we use various denoising algorithms to denoise the event stream after injecting noise to test and record the effect comparison of different algorithms and the effect changes of the same algorithm. The simulated event streams, in which the injected noise accounts for 20%, 50%, and 90% of the original event stream, and the diagrams before and after denoising are shown in Figure 8. The binary images in the figure are obtained by projecting the events within a period of time into the elements in the matrix corresponding to the position of the camera focal plane. These images serve as two-dimensional diagrams to intuitively display the performance differences of various algorithms. It should be emphasized that all the algorithms mentioned in the text are based on the asynchronous event stream itself for denoising processing, rather than on the binary plane image.

After processing the simulated event stream with each denoising algorithm, in order to quantitatively evaluate the results of denoising, we propose five evaluation indicators.

#### 3.1.1. True Rate of Events (ETR)

This index represents the proportion of true events remaining in the event stream after denoising and is denoted as ETR,
(18)$$ETR=\frac{S_{denoised\;true}}{S_{denoised}}.$$

#### 3.1.2. Loss Rate of Events (ELR)

This index represents the proportion of true events that are treated as noise and deleted thus lost in the total events after denoising; it is denoted as ELR,
(19)$$ELR=1-\frac{S_{denoised\;true}}{S_{total\;true}}.$$

#### 3.1.3. True Positive Rate of Events (ETPR)

This index represents the proportion of true events in the event stream after denoising to the total number of true events before denoising; it is denoted as ETPR. This index is also known as the true prediction rate,
(20)$$ETPR=\frac{S_{denoised\;true}}{S_{total\;true}}.$$

#### 3.1.4. False Positive Rate of Events (EFPR)

This index represents the proportion of noise events in the event stream after denoising to the total number of noise events before denoising; it is denoted as EFPR. This index is also known as the false omission rate,
(21)$$EFPR=\frac{S_{denoised\;noise}}{S_{total\;noise}}.$$

#### 3.1.5. Signal-to-Noise Ratio of Events (ESNR)

This index represents the ratio of the number of events correctly predicted as true events to the number of events incorrectly predicted as true events (that is, noise events treated as true events by the denoising algorithm) in the event stream after denoising; it is denoted as ESNR,
(22)$$ESNR=\frac{S_{denoised\;true}}{S_{denoised\;noise}}.$$

The variables in these indices represent the following: $S_{denoised\;true}$ is the number of true events in the event stream after denoising; $S_{denoised\;noise}$ is the number of noise events in the event stream after denoising; $S_{denoised}$ is the total number of events remaining after denoising; $S_{total\;true}$ is the total number of true events in the event stream before denoising; and $S_{total\;noise}$ is the total number of noise events in the event stream before denoising.

If the performance of the denoising algorithm is excellent, then the corresponding ETR, ETPR, and ESNR values will be higher, and the corresponding ELR and EFPR values will be lower. The statistical data of the experimental results are shown in the following Table 1, Table 2, Table 3, Table 4 and Table 5 and Figure 9, Figure 10 and Figure 11.

### 3.2. Denoising Experiment in Real Scene

The denoising effect on simulated event streams cannot fully represent the actual denoising capability of the algorithm. During actual shooting, in addition to background activity noise, there are various complex noises in the environment that cannot be simulated and predicted, including diffuse reflection noise. Thus, the removal effect of this kind of noise is one of the important indicators for evaluating the effect of the algorithm and is also one of the main focuses of our algorithm. Given that noise in the simulated event stream does not include these unconventional types, we need to use real event stream datasets to test the actual performance and effect of each algorithm.

We divide the filming situation into two cases: 1. the subject is stationary, the camera moves; and 2. the subject moves, the camera is stationary. Moreover, in the latter filming situation, we also shot a dataset of event streams containing a large area of environmental brightness changes due to the subject passing through a high-brightness light source, causing a large range of diffuse reflection noise. The two-dimensional diagrams of the dataset before and after being processing with each denoising algorithm are shown in Figure 12.

In Figure 12, Scene A and Scene B correspond to the filming situation where the camera is stationary and the subject moves, while Scene C corresponds to the filming situation where the camera moves and the subject is stationary. Among them, the denoising effect comparison diagram corresponding to Scene B shows the filming effect of the event camera when the environmental illumination information changes abruptly during the filming process, as well as the processing effects of each denoising algorithm. Through this set of experimental effect comparison charts, it can be intuitively felt that our algorithm not only has a better denoising effect when dealing with diffuse reflection noise, but also has a better ability to retain real events.

The filming effect of the event stream corresponding to Scene B before and after the abrupt change in illumination is shown in Figure 13. We can see that during this process, there were two large areas of unconventional noise, which occurred when the subject blocked and left the light source, respectively. The effect comparison diagram of Scene B was chosen when the subject left the light source. From the figure we can see that the frame images obtained with the APS channel of the DAVIS346 camera used in the experiment lost the contour information of the human back, which was caused by the overexposure of the photosensitive device. However, the event stream obtained with the DVS channel still contains the contour information of the human back, which reflects the high dynamic range advantage of the event camera.

Next, we need to make a quantitative comparison of the datasets processed with each algorithm. However, when dealing with real event stream data, we cannot know in advance which are the real events and which are the noise events like simulated event streams. Therefore, new evaluation criteria are needed to compare the denoising effects of each algorithm. Here, we propose two evaluation indicators.

#### 3.2.1. Signal-to-Noise Ratio of Events ($p_{SNR}$)

This index is different from the ESNR used in the denoising experiment in the simulated scene. It is based on the traditional formula for evaluating image quality signal-to-noise ratio, and then modified according to the characteristics of the event stream. This index is the ratio of the denoised event stream quantity to the number of events treated as noise and deleted after denoising, denoted as $p_{SNR}$, which can be used to represent the quality of the event stream,
(23)$$p_{SNR}\;=\;10\log_{10}\left(\frac{S_{signal}}{S_{noise}}\right),$$

$S_{signal}$ represents the total number of events in the event stream after denoising; and $S_{noise}$ represents the number of events that are treated as noise and removed after denoising.

#### 3.2.2. Dynamic Phase Signal Performance Indicator ($p_{DPSPI}$)

The $p_{SNR}$, which is improved based on the traditional signal-to-noise ratio formula, is a result-oriented evaluation indicator and is indispensable when evaluating algorithm performance. However, in the formula, we do not know whether the noise represented by $S_{noise}$ contains real events that are misidentified as noise or whether the event stream output after denoising contains noise that is misidentified as a real event. Therefore, we define a new indicator $p_{DPSPI}$, which quantitatively evaluates the quality of preserving real events during the denoising process,
(24)$$p_{DPSPI}=\frac{\sum_{t=0}^{T}t(u_{T}+w_{T}+v_{T})e^{\frac{10}{\tau \pi }[sin(u_{T}+w_{T}+v_{T}){]}^{2}}}{T^{3}+1}.$$

In the discrete integration process corresponding to the numerator of the formula (24), the relative timestamp t where an event occurred within [0, T] will be included in the calculation. The parameters involved are defined as follows:(25)$$u_{T}=10\;\log_{10}\left(t\sqrt{{\left(x_{T}-{\tilde{x}}_{T}\right)}^{2}}+1\right),$$
(26)$$w_{T}=10\;\log_{10}\left(t\sqrt{{\left(y_{T}-{\tilde{y}}_{T}\right)}^{2}}+1\right),$$
(27)$$v_{T}=5\;\log_{10}\left[t\sqrt{{\left(x_{T}-{\tilde{x}}_{T}\right)}^{2}+{\left(y_{T}-{\tilde{y}}_{T}\right)}^{2}}+1\right],$$
(28)$$\tau =\sqrt[{6}]{{\left(\frac{x_{T}-{\tilde{x}}_{T}}{y_{T}-{\tilde{y}}_{T}}\right)}^{2}}.$$

$x_{T}$ represents the weighted average of the horizontal coordinates of the pixel locations where events have occurred in the dataset after denoising within the time period [T − 5000, T], with the frequency of events occurring at a certain pixel location used as the weight. ${\tilde{x}}_{T}$ represents the weighted average of the horizontal coordinates of the pixel locations where events have occurred in the dataset before denoising within the time period [T − 5000, T]. The unit of T is microseconds (μs).

Similarly, $y_{T}$ represents the weighted average of the vertical coordinates of the pixel locations where events have occurred in the dataset after denoising within the time period [T − 5000, T]. ${\tilde{y}}_{T}$ represents the weighted average of the vertical coordinates of the pixel locations where events have occurred in the dataset before denoising within the time period [T − 5000, T].

As T increases, if the denoising algorithm has a good effect on preserving real events, $p_{DPSPI}$ is always in a convergent state, that is, the calculation result decreases as time T increases. If $p_{DPSPI}$ starts to diverge at a certain time point T_a_ and the calculation result starts to reconverge at time T_b_, then the denoising program does not have an ideal effect on preserving real events within the corresponding time range (T_a_,T_b_), and removes too much event data that should be preserved.

The corresponding event stream data time lengths for Scene A, Scene B, and Scene C are 3,874,933 μs, 7,029,959 μs, and 5,279,061 μs, respectively. When conducting denoising experiments and recording data for the two indicators, we first divide the event stream data before and after denoising into data segments every 5 ms according to the order of timestamp changes, then calculate and record the $p_{SNR}$ within each segment. For the $p_{DPSPI}$ indicator, we also record data every 5 ms. That is, each time we increase the upper limit T in the formula, we increase its size by a change of 5000 (which is 5 ms), and then recalculate the size of $p_{DPSPI}$ within [0, T] and record it.

If the $p_{SNR}$ corresponding to the denoising algorithm is smaller, it indicates that its denoising performance in the conventional sense is more excellent; when the declining trend of $p_{DPSPI}$ corresponding to the denoising algorithm is more significant and the rising trend is less noticeable, then the preservation effect of real event data is better. In other words, when the derivative of $p_{DPSPI}$ is negative and the value is larger, the effect is better. Let $\sigma_{1}$ represent the average value of $p_{SNR}$ and let $\sigma_{2}$ represent the average value of the derivative of $p_{DPSPI}$; then, the larger the −$(\sigma_{2}$/$\sigma_{1})$, the better the combined ability of the corresponding denoising program to remove noise events and retain real events. The statistical data of the experimental results are shown in the Table 6 and Figure 14 below:

### 3.3. Ablation Experiment

In this section, we conducted an ablation experiment to investigate the role of the MPAR algorithm in the entire denoising process. We continue to use the same data and metrics as in the denoising experiment in real scene. Instead of employing the MPAR algorithm to divide the event stream into the main event stream U and the non-main event stream N during the denoising process, we directly treat the entire event stream as the main event stream U. We then quantify the relationship between events using the energy function and apply the EEIM algorithm for denoising. The comparison of experimental results and corresponding data statistics with and without using the MPAR algorithm are presented in Table 7 and Figure 15 and Figure 16.

## 4. Discussion

The experimental part of this paper fully demonstrates that our denoising algorithm maintains excellent comprehensive denoising capabilities when dealing with different noise and shooting environments. In the simulated scenario of the experimental stage, we add random noise that conforms to the Poisson distribution to the event stream, simulating the main categories of noise encountered in most shooting environments. Whether it is the event stream generated via the event camera simulator or the noise added later, we can know their accurate coordinates and the time nodes of their occurrence. The advantage of doing this is that we can arbitrarily modify the amount of added noise, thereby testing the robustness of the denoising algorithm when the noise density changes. When using various algorithms to denoise the event stream that has added noise, we can clearly know the retention situation of each real event and noise event, and then use five indicators to quantify it into various aspects of event stream processing performance. Judging from the statistical results of the data, our denoising method has the highest values for ETR, ETPR, and ESNR and the lowest values for ETR and EFPR. This means that our denoising algorithm not only has a better noise removal effect but also has a better preservation effect on real events during the denoising process. Moreover, as the noise gradually increases, our algorithm maintains the ability to keep the denoising effect stable, that is, the robustness of the algorithm is relatively stronger than other denoising algorithms.

Next is the real-scene denoising experiment conducted to make up for the insufficient factors involved in the simulation scenario. From the statistical results of the data, our algorithm’s $\sigma_{2}$ in Scene A is slightly higher than the $\sigma_{2}$ corresponding to Guo denoising algorithm, $\sigma_{2}$ in Scene B of ours is almost the same as $\sigma_{2}$ corresponding to Yang denoising algorithm, and $\sigma_{1}$ in Scene C is almost the same as $\sigma_{1}$ corresponding to Guo denoising algorithm. However, to evaluate the performance of the denoising algorithm, we need to consider two indicators together, because when too many events are removed the real events lost will also increase; when too many events are retained, the noise events that have not been removed will also increase. The value of −$\left({\sigma_{2}/\sigma }_{1}\right)$ corresponding to our algorithm is always the highest. This shows that the comprehensive denoising ability of our algorithm is significantly better than other algorithms. In the ablation experiment, we observed the role of the MPAR algorithm in the entire denoising process by enabling and disabling it. The experimental results show that without the MPAR algorithm, our method still performs well in retaining real events; however, its ability to remove noise is significantly reduced, especially when dealing with unconventional noises such as diffuse reflection noise.

## 5. Conclusions

This paper proposes a denoising method for event streams based on main body area recognition. By using the MPAR algorithm, which distinguishes the main body event stream from the non-main body event stream based on the overall correlation of the event stream, we can distinguish irregular noise, such as diffuse reflection noise that has a strong correlation between events. Then, through subsequent steps of quantifying the relationship between events using an energy function, and considering the degree of correlation in time and space between events within different event streams and the same event stream using the EEIM algorithm, we can more accurately distinguish noise events that have a strong or weak correlation with real events. In this way, we can obtain an event stream that has removed most regular and irregular noises and better preserved the real events. To verify the effectiveness of this algorithm, we conducted denoising experiments with simulated event stream datasets and real event stream datasets during the comparison experimental stage. Moreover, in the real scene denoising stage, we proposed a new evaluation method to quantitatively evaluate the preservation of real events in the real scene event stream after denoising. The experimental results show that our algorithm has superior comprehensive performance.

## Figures and Tables

**Figure 1 sensors-23-06655-f001:**
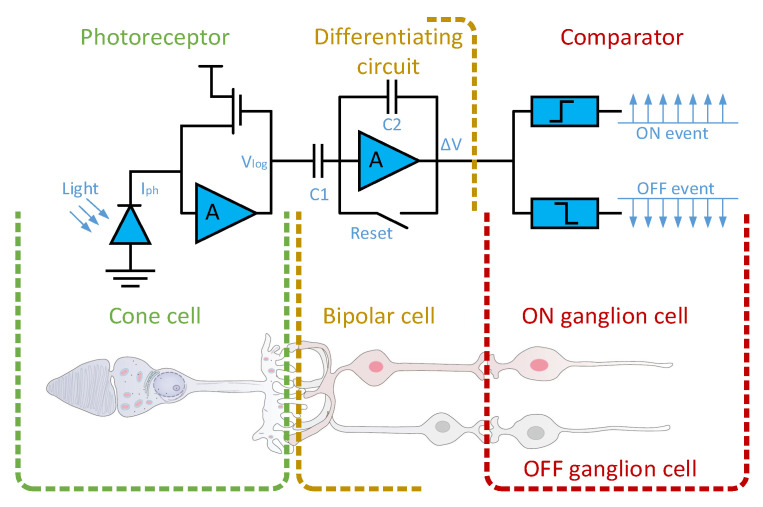
Three-layer model of a human retina and corresponding DVS pixel circuitry. The first layer is similar to retinal cone cells for photoelectric conversion; the second layer, similar to bipolar cells in the retina, is used to obtain changes in light intensity; the third layer is similar to the ganglion cells of the retina for outputting the light intensity change sign.

**Figure 2 sensors-23-06655-f002:**
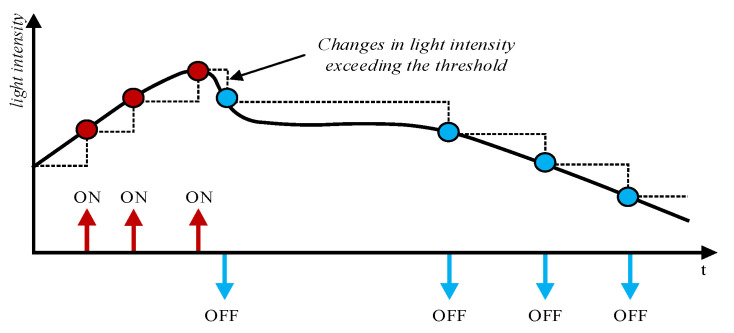
The generation process of an event. An ON event occurs when the voltage rises and the change exceeds the threshold. An OFF event occurs when the voltage drops and the change exceeds the threshold.

**Figure 3 sensors-23-06655-f003:**
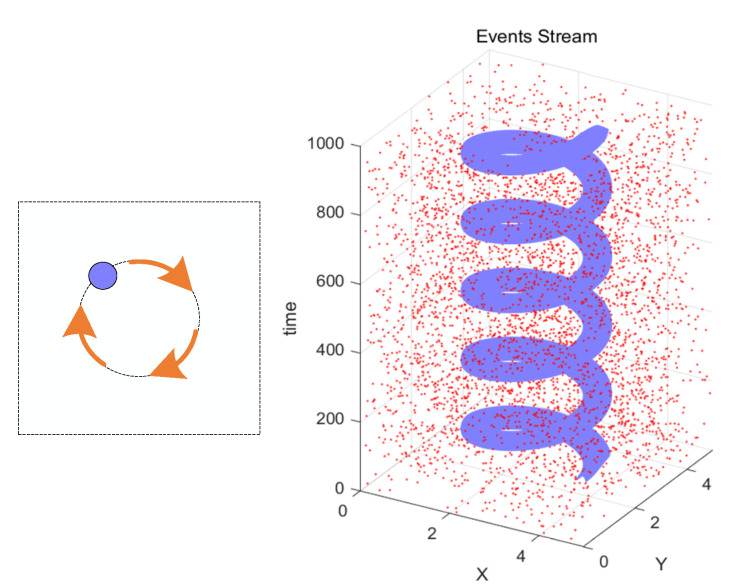
The event stream captured via the event camera when a ball rotates around a point. The red dots in the figure represent noise events, while the purple dots represent real events.

**Figure 4 sensors-23-06655-f004:**
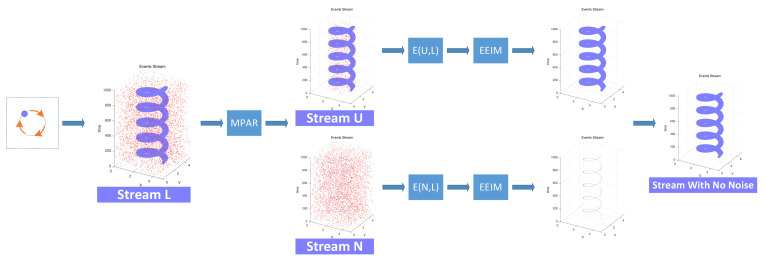
The complete process of denoising the event stream. The red dots in the figure represent noise events, while the purple dots represent real events.

**Figure 5 sensors-23-06655-f005:**
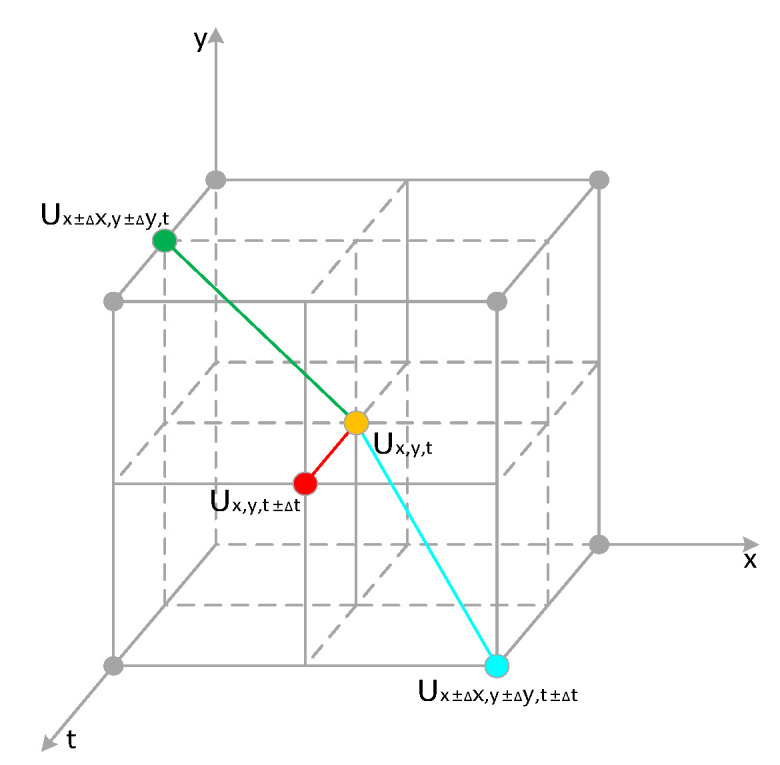
Spatial and temporal correlation between events.

**Figure 6 sensors-23-06655-f006:**
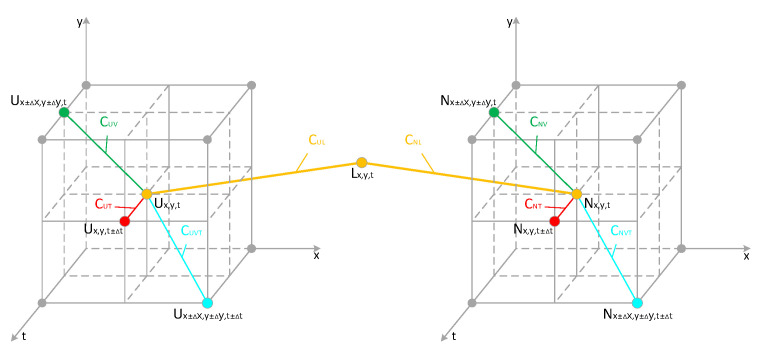
The maximum clique corresponding to each energy function.

**Figure 7 sensors-23-06655-f007:**
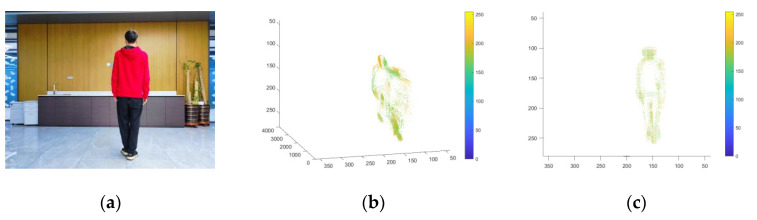
(**a**) Original high-frame-rate video material. (**b**,**c**) are diagrams of the event stream simulated from high-frame-rate videos of the event camera simulator.

**Figure 8 sensors-23-06655-f008:**
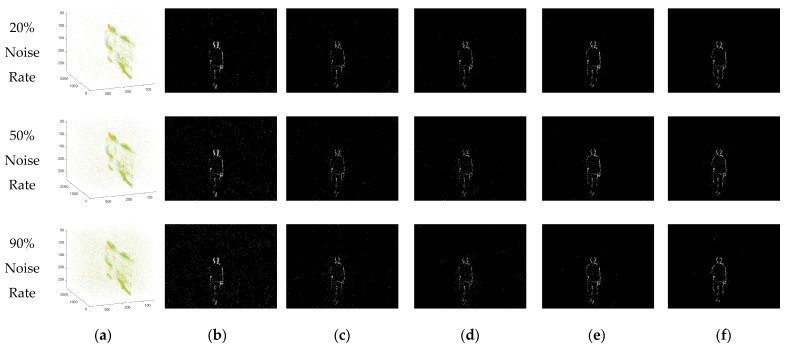
(**a**) Three-dimensional visualization of the event stream after adding noise of different proportions; (**b**) two-dimensional visualization of the original event stream; (**c**) the denoising result processed via the Khodamoradi algorithm; (**d**) the denoising result processed via the Yang algorithm; (**e**) the denoising result processed via the Guo algorithm; (**f**) the denoising result processed via our algorithm.

**Figure 9 sensors-23-06655-f009:**
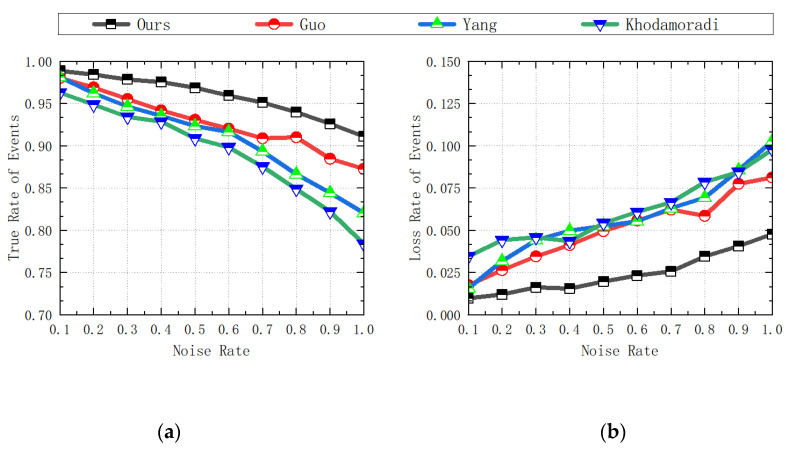
Comparison of ETR and ELR data for each algorithm: (**a**) ETR data; (**b**) ELR data [31,36,37].

**Figure 10 sensors-23-06655-f010:**
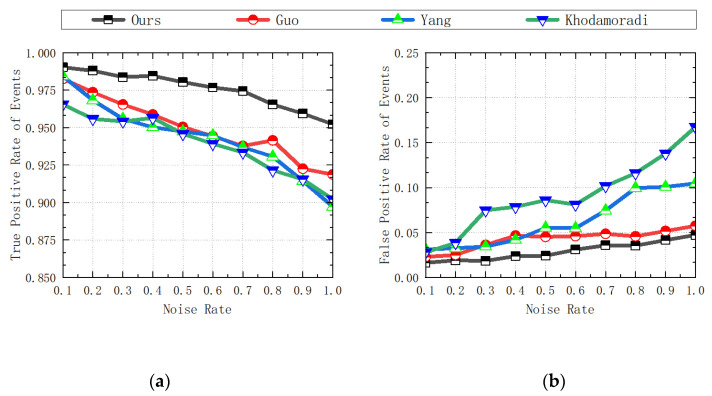
Comparison of ETPR and EFPR data for each algorithm: (**a**) ETPR data; (**b**) EFPR data [31,36,37].

**Figure 11 sensors-23-06655-f011:**
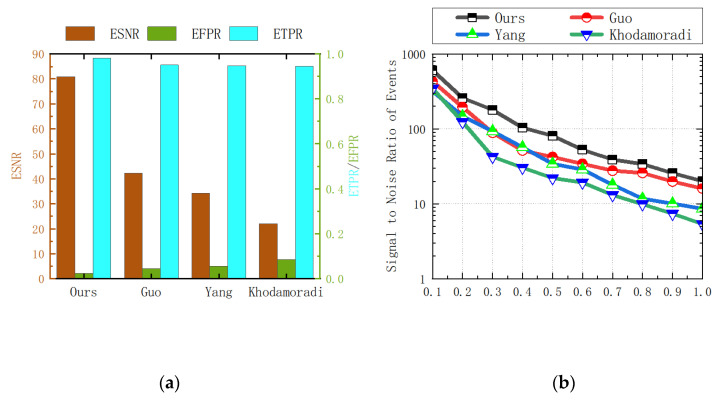
(**a**) When the noise ratio is 50%, comparison of ETPR, EFPR, and ESNR data statistics results for each algorithm; (**b**) comparison of ESNR data for each algorithm [31,36,37].

**Figure 12 sensors-23-06655-f012:**
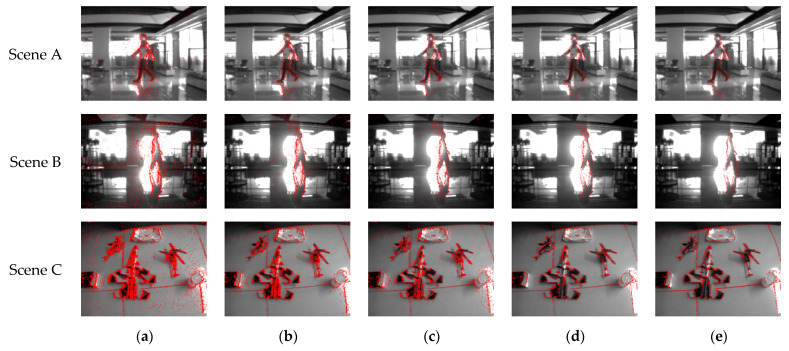
(**a**) Two-dimensional visualization of the original event stream; (**b**) the denoising result processed via the Khodamoradi algorithm; (**c**) the denoising result processed via the Yang algorithm; (**d**) the denoising result processed via the Guo algorithm; (**e**) the denoising result processed with our algorithm.

**Figure 13 sensors-23-06655-f013:**
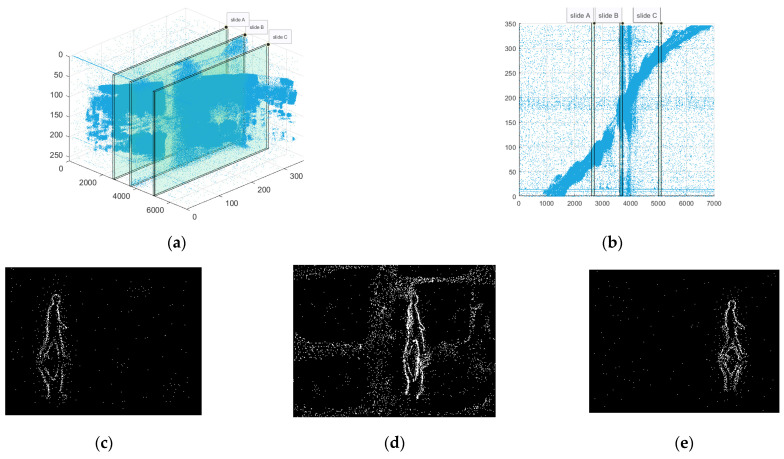
(**a**,**b**) are three-dimensional visualization of the original event stream, and three sections of the event stream projected to the focal plane in chronological order, which are slide A, slide B, and slide C; (**c**) slide B; (**d**) slide A; and (**e**) slide C.

**Figure 14 sensors-23-06655-f014:**
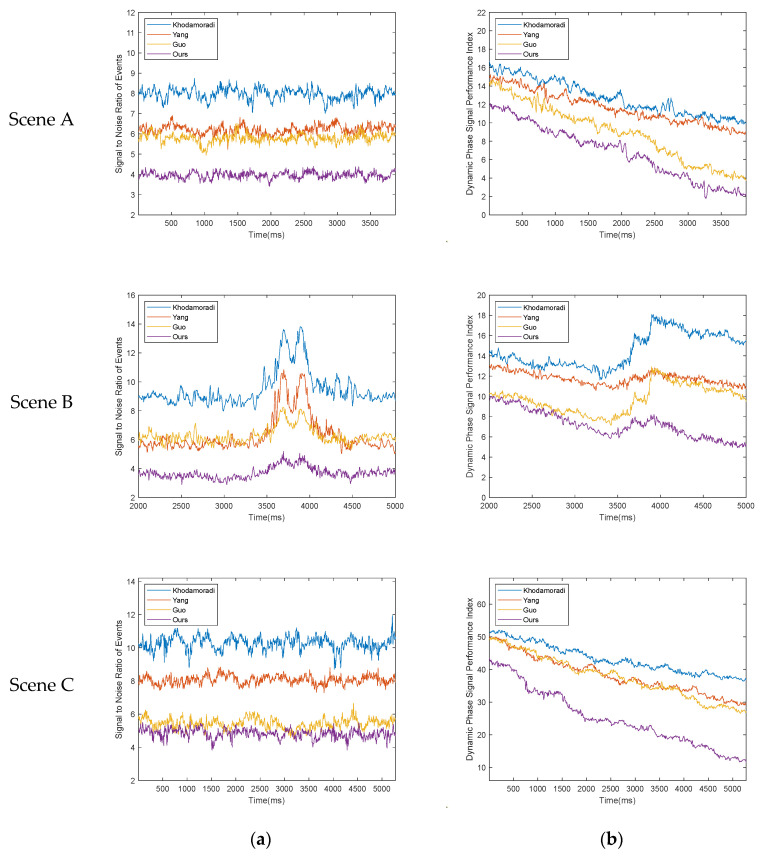
The corresponding $p_{SNR}$ and $p_{DPSPI}$ after each denoising algorithm has processed the event stream dataset under three shooting scenarios: (**a**) $p_{SNR}$ data statistics for each algorithm under three scenarios; (**b**) $p_{DPSPI}$ data statistics for each algorithm under three scenarios.

**Figure 15 sensors-23-06655-f015:**
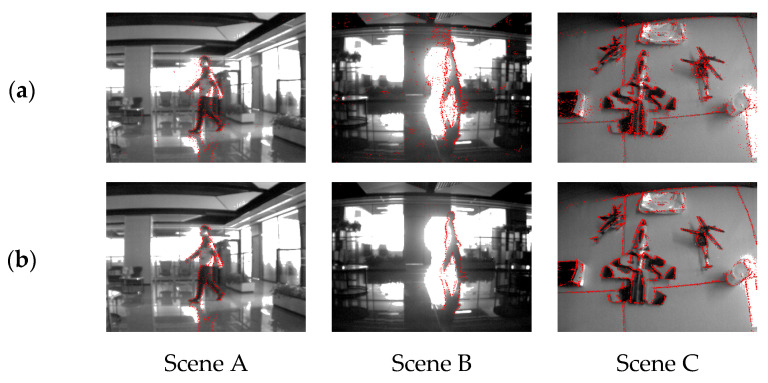
(**a**) The denoising result processed with our algorithm without using the MPAR; (**b**) the denoising result processed with our algorithm using the MPAR.

**Figure 16 sensors-23-06655-f016:**
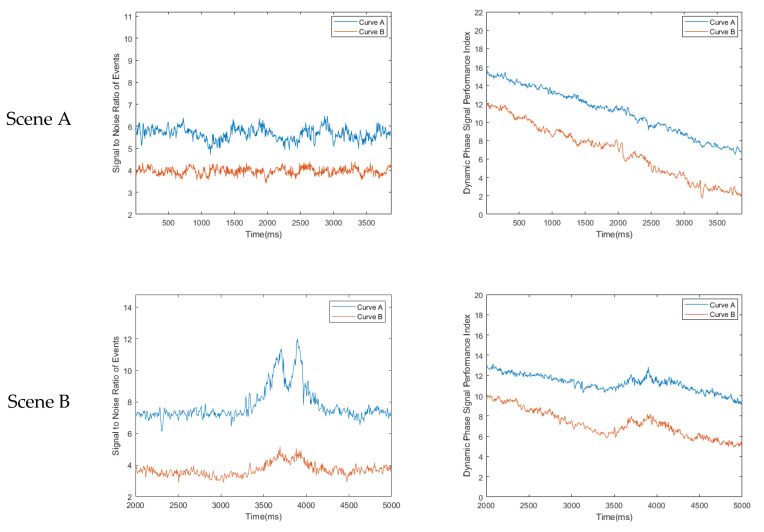
The corresponding $p_{SNR}$ and $p_{DPSPI}$ after two types of algorithms have processed the event stream dataset: (**a**) $p_{SNR}$ data statistics under three scenarios; (**b**) $p_{DPSPI}$ data statistics under three scenarios. Curve A corresponds to the algorithm without using the MPAR. Curve B corresponds to the algorithm with using the MPAR.

**Table 1 sensors-23-06655-t001:** ETR data statistics for each algorithm.

Method	True Rate of Events, ETR									
	10%	20%	30%	40%	50%	60%	70%	80%	90%	100%
Ours	98.86%	98.42%	97.85%	97.54%	96.88%	95.94%	95.12%	93.98%	92.62%	91.12%
Guo [37]	98.03%	96.89%	95.54%	94.19%	93.06%	92.02%	90.89%	91.02%	88.46%	87.27%
Yang [36]	98.13%	96.21%	94.65%	93.54%	92.32%	91.63%	89.32%	86.65%	84.44%	82.03%
Khodamoradi [31]	96.28%	94.88%	93.42%	92.85%	90.89%	89.81%	87.54%	84.89%	82.21%	78.38%

**Table 2 sensors-23-06655-t002:** ELR data statistics for each algorithm.

Method	Loss Rate of Events, ELR									
	10%	20%	30%	40%	50%	60%	70%	80%	90%	100%
Ours	0.98%	1.21%	1.62%	1.55%	1.97%	2.32%	2.56%	3.45%	4.06%	4.78%
Guo [37]	1.75%	2.64%	3.46%	4.13%	4.95%	5.58%	6.22%	5.85%	7.75%	8.13%
Yang [36]	1.57%	3.18%	4.42%	4.98%	5.26%	5.52%	6.31%	6.94%	8.56%	10.28%
Khodamoradi [31]	3.46%	4.42%	4.58%	4.36%	5.42%	6.09%	6.65%	7.85%	8.46%	9.78%

**Table 3 sensors-23-06655-t003:** ETPR data statistics for each algorithm.

Method	True Positive Rate of Events, ETPR									
	10%	20%	30%	40%	50%	60%	70%	80%	90%	100%
Ours	99.02%	98.79%	98.38%	98.45%	98.03%	97.68%	97.44%	96.55%	95.94%	95.22%
Guo [37]	98.25%	97.36%	96.54%	95.87%	95.05%	94.42%	93.78%	94.15%	92.25%	91.87%
Yang [36]	98.43%	96.82%	95.58%	95.02%	94.74%	94.48%	93.69%	93.06%	91.44%	89.72%
Khodamoradi [31]	96.54%	95.58%	95.42%	95.64%	94.58%	93.91%	93.35%	92.15%	91.54%	90.22%

**Table 4 sensors-23-06655-t004:** EFPR data statistics for each algorithm.

Method	False Positive Rate of Events, EFPR									
	10%	20%	30%	40%	50%	60%	70%	80%	90%	100%
Ours	1.63%	1.90%	1.84%	2.37%	2.42%	3.09%	3.58%	3.54%	4.15%	4.73%
Guo [37]	2.28%	2.49%	3.61%	4.65%	4.50%	4.60%	4.84%	4.57%	5.16%	5.74%
Yang [36]	3.11%	3.27%	3.43%	4.16%	5.53%	5.49%	7.46%	9.94%	10.07%	10.45%
Khodamoradi [31]	2.80%	3.86%	7.48%	7.85%	8.59%	8.10%	10.16%	11.60%	13.78%	16.74%

**Table 5 sensors-23-06655-t005:** ESNR data statistics for each algorithm.

Method	Signal-to-Noise Ratio of Events, ESNR									
	10%	20%	30%	40%	50%	60%	70%	80%	90%	100%
Ours	605.824	259.602	178.689	103.890	80.957	52.609	38.928	34.088	25.678	20.151
Guo [37]	430.132	195.408	89.043	51.530	42.249	34.182	27.659	25.777	19.863	16.012
Yang [36]	316.910	147.850	92.976	57.064	34.241	28.699	17.941	11.707	10.086	8.587
Khodamoradi [31]	345.126	123.826	42.529	30.441	22.034	19.318	13.130	9.929	7.384	5.388

**Table 6 sensors-23-06655-t006:** The values of $\sigma_{1}$ and $\sigma_{2}$ corresponding to the event dataset in three shooting scenarios after being processed with various noise reduction algorithms.

Method	Scene A			Scene B			Scene C		
	$\sigma_{1}$	$\sigma_{2}$	$-(\sigma_{2}$ $/\sigma_{1})$	$\sigma_{1}$	$\sigma_{2}$	$-(\sigma_{2}$ $/\sigma_{1})$	$\sigma_{1}$	$\sigma_{2}$	$-(\sigma_{2}$ $/\sigma_{1})$
Ours	3.96	−2.61 × 10^−3^	6.59 × 10^−4^	3.71	−1.38 × 10^−3^	3.71 × 10^−4^	4.83	−5.56 × 10^−3^	1.15 × 10^−3^
Guo [37]	5.79	−2.74 × 10^−3^	4.73 × 10^−4^	6.30	6.87 × 10^−4^	−1.09 × 10^−4^	5.44	−4.28 × 10^−3^	7.87 × 10^−4^
Yang [36]	6.21	−1.56 × 10^−3^	2.51 × 10^−4^	6.37	−3.65 × 10^−4^	5.73 × 10^−5^	8.06	−3.66 × 10^−3^	4.54 × 10^−4^
Khodamoradi [31]	7.97	−1.65 × 10^−3^	2.07 × 10^−4^	9.52	1.35 × 10^−3^	−1.42 × 10^−4^	10.27	−2.86 × 10^−3^	2.79 × 10^−4^

**Table 7 sensors-23-06655-t007:** The values of $\sigma_{1}$ and $\sigma_{2}$ corresponding to the event dataset in three shooting scenarios after being processed with two types of algorithms.

Method	Scene A			Scene B			Scene C		
	$\sigma_{1}$	$\sigma_{2}$	$-(\sigma_{2}$ $/\sigma_{1})$	$\sigma_{1}$	$\sigma_{2}$	$-(\sigma_{2}$ $/\sigma_{1})$	$\sigma_{1}$	$\sigma_{2}$	$-(\sigma_{2}$ $/\sigma_{1})$
without MPAR	5.66	−2.28 × 10^−3^	4.03 × 10^−4^	7.82	−7.91 × 10^−4^	1.01 × 10^−4^	6.18	−3.87 × 10^−3^	6.34 × 10^−4^
with MPAR	3.96	−2.61 × 10^−3^	6.59 × 10^−4^	3.71	−1.38 × 10^−3^	3.71 × 10^−4^	4.83	−5.56 × 10^−3^	1.15 × 10^−3^

## Data Availability

Not applicable.

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
