# Peer review of "Denoising Method Based on Salient Region Recognition for the Spatiotemporal Event Stream"

_sensors, 2023, doi:10.3390/s23156655_

Round 1
Reviewer 1 Report
The authors submitted an interesting and a well written manuscript that fits with the scope of the journal and the subject analysed is of great interest. The methodology is well described and the conclusions are supported by the results. However, the authors should take into account all the comments and the suggestions from reviewers and editors before the paper could be considered for publication.
Minor editing of English language required
Author Response
Dear Reviewer,
Thank you for taking the time to review our paper 《Denoising Method Based on Salient Region Recognition for the Spatiotemporal Event Stream》 and for providing valuable feedback and suggestions. We wish you a pleasant life and success in your work.
Sincerely,
Tang Sichao
Reviewer 2 Report
The manuscript describes an event stream denoising method based on salient region recognition. The manuscrip requires some revisions before it can be published.
1) Line 214-226: Since the authors have proposed several novel designs in this work, a further ablation analysis in the experimental part is required to demonstrate their advantages separately.
Be specific, say there are three novel designs, A, B, and C. The authors are required to test if only working eliminating the part A, B, or C, what would the results be, in order to demonstrate the advantage of each novel design.
2) Algorithm 1 and 2: Please try not to use too many plain languages in algorithms. Instead, use pseudo code and include any mathematical details for clarity.
Language is fine but minor editing is required.
Author Response
Dear Reviewer,
Thank you for taking the time to review our paper 《Denoising Method Based on Salient Region Recognition for the Spatiotemporal Event Stream》 and for providing valuable feedback and suggestions. We have carefully read your comments and made corresponding revisions based on the issues and suggestions you raised. Below, we respond to and clarify your concerns, and we hope these revisions meet your requirements.
Point 1:“Line 214-226: Since the authors have proposed several novel designs in this work, a further ablation analysis in the experimental part is required to demonstrate their advantages separately.
Be specific, say there are three novel designs, A, B, and C. The authors are required to test if only working eliminating the part A, B, or C, what would the results be, in order to demonstrate the advantage of each novel design. ”
Response 1: We have added an ablation study for the MPAR algorithm. The role of the MPAR algorithm is to pre-classify the event stream into two event streams, each containing both real events and noise events. If only the MPAR algorithm is used, we would obtain two event streams with incomplete information, and noise events would still be present, merely distributed across these two streams. The specific steps to distinguish and remove noise events within each event stream are carried out in the subsequent stages, including the step that employs the energy function to quantify the relationships between events and the step corresponding to the EEIM algorithm. Therefore, although the step corresponding to the MPAR algorithm can be removed, the subsequent algorithm steps are indispensable. In our experiments, we observed the role of the MPAR algorithm in the entire denoising process by enabling and disabling it to obtain different experimental results. The specific ablation study section corresponds to lines 668-684 and 717-721 in the Newly uploaded paper paper.
Point 2:“Algorithm 1 and 2: Please try not to use too many plain languages in algorithms. Instead, use pseudo code and include any mathematical details for clarity.”
Response 2: We have have tired our best to remove redundant plain language and streamline the process in Algorithm 1 and 2, adding some mathematical details where appropriate.
Please review our revised paper again. If you have any questions or need further clarification, feel free to let us know. We appreciate your attention to and review of our work.
Sincerely,
Tang Sichao
Round 2
Reviewer 2 Report
The reviewer thanks the authors to make such changes. The revised manuscript looks better and it has addressed my comments.
The revised manuscript is now recommended as acceptance as it is.
Language is generally fine but minor editing (proof) is required.